# Determinants of stroke among adult hypertension patients in dessie comprehensive specialized hospital and Kombolcha general hospital, Amhara, Ethiopia: A case-control study

**Muluken Demile[1], Molla Kassie Yenesew[2], Abebe Bekele Jibat[3], Yohannes Kibret[4], Dagnachew Melak[5], Endalew Minwuye Andargie[6], Yonas Fissha Adem** [4]*

**1** Kombolcha 03 health center, Kombolcha, Ethiopia, **2** Family Guidance Association of Ethiopia, Woldia, Ethiopia, **3** Department of Nursing, Tropical College of Medicine, Dessie, Ethiopia, **4** Department of Public Health, Dessie College of Health Sciences, Dessie, Ethiopia, **5** Department of Epidemiology and Biostatistics, Wollo University, Dessie, Ethiopia, **6** Department of Public Health, School of Public Health, Asrat Woldeyes Health Science Campus, Debre Berhan University, Debre Berhan, Ethiopia

* yonasfissha029@gmail.com

## Abstract

### Introduction

Despite the increased admission of stroke patients in hospitals as a result of hypertension, especially in Ethiopia, there are limited studies that aim to explore important determinants of stroke, mainly among adult hypertensive patients. Therefore, this study was conducted to identify determinants of stroke among adult hypertension patients in Dessie Comprehensive Specialized Hospital and Kombolcha General Hospital, Amhara, Ethiopia.

### Methods

An institutional-based case-control study design was conducted at Dessie Comprehensive Specialized Hospital and Kombolcha General Hospital from April 10, 2023, to May 30, 2023, on 105 cases and 420 controls selected by using a systematic random sampling technique. Data entry and processing were done using EPI-data version 4.4.2 and exported to SPSS version 26.0 for further analysis. Bi-variable and multi-variable logistic analysis was done to identify determinants of stroke among hypertension patients. Data were interpreted by using the odds ratio with a 95% confidence level, and a P-value < 0.05 was taken as statistically significant.

### Results

A total of 105 cases and 420 controls participated in this study with a complete response (100%). Stroke among adult hypertension patients was significantly associated with a family history of hypertension (AOR = 2.8, 95% CI: 1.8–4.3), being physically inactive (AOR = 2.13, 95% CI: 1.4–3.3), current smoking cigarettes (AOR = 3.2, 95% CI: 2.1–5.2), current

**Data availability statement:** All relevant data are within the paper and its Supporting Information files.

**Funding:** The author(s) received no specific funding for this work.

**Competing interests:** The authors have declared that no competing interests exist.

alcohol drinking (AOR = 4.7, 95% CI: 2.87–7.9), and having heart disease (AOR = 2.04, 95% CI: 1.3–3.25).

## Conclusions

Having a family history of hypertension, cigarette smoking, physical inactivity, alcohol consumption, and heart disease were determinants of stroke. Health education on lifestyle practices, early hypertension, and other cardiovascular diseases in each follow-up visit is very essential for improving stroke prevention.

## Introduction

Stroke, or cerebrovascular accident, is one of the non-communicable diseases that is defined as the abrupt onset of a neurologic deficit that is caused by the interruption of the blood supply to the brain and results in cell death [1–3]. The burden of stroke is increasing at an alarming rate globally, the second most common cause of mortality worldwide, comprising approximately 10% of all deaths and killing 5.5 million people each year, with 44 million disability-adjusted life years (DALYs) lost [4]. Of all, 75.2% of stroke mortality and 81.0% of stroke-related disability-adjusted life years were in developing countries, which is about 3.8 times higher than in developed countries [5]. Globally, over 23 and 7.8 million new stroke cases and deaths are expected by 2030, respectively, if there is no intervention. Besides, it remains the leading cause of disability among adults [6–8]. Stroke patients and society at large bear a heavy financial burden due to the significant demands placed on family members and caregivers [9].

The most frequent risk factor for stroke is hypertension, which is also the primary modifiable factor linked to stroke-related disability [10,11]. Stroke risk rises seven times with untreated hypertension [12]. Seventy percent of individuals with intracerebral hemorrhage had hypertension [12–14]. Distinct forms of hypertension are linked to distinct kinds of stroke. The risk of stroke and ischemic heart disease could be lowered by 40% and 14%, respectively, for every 5 mm Hg decrease in diastolic blood pressure (DBP), according to a previous antihypertensive therapy trial that used DBP (≥90 mmHg) as the inclusion criterion [12]. Globally, one in three adults has hypertension, a condition which causes 51% of all deaths from stroke [15]. Better control of hypertension and its associated risk factors has decreased the burden of stroke in high-income countries (HIC) [15,16]. However, in Africa, where hypertension was once rare, more than half of the adult population in some countries is estimated to be hypertensive and the burden is increasing at alarming proportions. Every minute, about six Africans suffer a stroke of which up to 98% may be hypertensive [15,17].

Different studies in different parts of the world have indicated that age, sex, smoking, limited physical exercise, obesity, alcohol consumption, non-adherence to antihypertensive medication, uncontrolled blood pressure, being diabetic, high cholesterol levels, and consuming unhealthy diets are the factors linked to stroke [1,2,8].

In Ethiopia, stroke is currently one of the greatest public health problems, accounting for 7% of total deaths. In addition to this, a study in northern Ethiopia showed that stroke was the third most common cause of medical intensive care unit admission (15.2%) and the first cause of death, accounting for 17% of all deaths in the medical intensive care unit [1,6]. Likewise, 38% of all strokes were brought on by antihypertensive medication, and 66.2% of all stroke admissions were caused by hypertension [2]. Another study, from north-west Ethiopia, showed a 69.4% ischemic stroke rate and a 13% hospital mortality rate [7]. In

2015/16–2019/20, according to the Ethiopian Health Sector Development Program, non-communicable diseases (NCDs) will cause a 12.5% drop in premature death [8]. According to recent WHO data, stroke deaths in Ethiopia reached 28,320, or 4.71% of total deaths. The age-adjusted death rate of 71.94 per 100,000 people ranks Ethiopia one hundred seventh in the world [1].

Furthermore, the Federal Ministry of Health in Ethiopia designed strategies to alleviate the problems encountered by non-communicable diseases, including stroke. According to the strategies, it can be prevented with lifestyle changes and controlling major risk factors, including hypertension, diabetes, and heart diseases [18,19]. Currently, community-based stroke prevention programs are one of the missing components at the ground level by using health extension workers, which was only stated in papers, and attention is focusing on communicable disease programs, putting the problem of stroke on the rise. Beyond this, while the number of stroke patients being admitted to hospitals as a result of hypertension is occasionally rising, few results attempt to investigate those factors through case-control studies since these few studies concentrate on the magnitude [7]. Furthermore, there is no similar study conducted in the study area, that accounts for differences in socioeconomic status, and access to treatment and it is useful for designing contextual interventions. Therefore, this study aims to assess the determinants of stroke among hypertensive patients at Dessie Comprehensive Specialized Hospital and Kombolcha General Hospital, Amhara, Ethiopia.

## Methods and materials

### Study design, geographic scope, and time frame

An institutional-based case-control study design was conducted from April 10, 2023, to May 30, 2023, in Dessie Comprehensive Specialized Hospital (DCSH) and Kombolcha General Hospital. Dessie Comprehensive Specialized Hospital (DCSH) is found in Dessie, Amhara regional state, Ethiopia. Kombolcha General Hospital is located in Kombolcha town, 20.9 km from Dessie. Dessie is found at 401 km north of Addis Ababa. The Dessie Comprehensive Specialized Hospital comprises four major departments and many specialist units, with a bed capacity of approximately 400. The patient flow at DCSH is above five hundred thousand per year. The hospital provides hypertensive services in the cardiac unit and stroke services in the neurology unit [20].

### Study population

All adult hypertensive patients and stroke patients from hypertension backgrounds in DCSH and Kombolcha General Hospital visit the facilities during the study period.

### Inclusion and exclusion criteria

**Inclusion criteria for cases.** All sampled adult stroke patients were from a hypertension background diagnosed by the neurologist (consultant internist) clinically or by confirmatory methods like CT-scan, and patients whose age was greater than 18 years, were included in the study.

**Inclusion criteria for controls.** The study included adult hypertension patients older than eighteen years who did not exhibit clinical signs of stroke or a history of stroke as controls.

**Exclusion criteria for cases.** Patients with fewer than three follow-ups for hypertension treatment before the first stroke occurrence.

**Exclusion criteria for controls.** Patients with a history of stroke or less than three follow-up appointments for treatment of hypertension were not included in the study.

## Sample size determination

The sample size calculation was done by double population proportion formulausing Epi Info statistical software, version 7.2.3.1, and the largest sample size was obtained by considering all the following assumptions: 1.81 odds ratio, a control-to-case ratio of 4:1, and 27.9% of controls among the exposed group and 41.2% of cases among the exposure with the eating fatty foods variable, from previous studies. There was an accepted error of 5%, a power of 80%, and a 95% confidence level. Adding the 10% non-response rate, the sample size obtained considering the above assumptions was 525 (105 for cases and 420 for controls). (S1 Table in S1 File)

## Sampling technique

A systematic random sample technique was employed to recruit both eligible cases and controls. Currently, at DCSH, 3496 hypertensive patients take follow-up antihypertensive drugs, and 910 patients have a stroke history, and at Kombolcha General Hospital, 782 hypertensive patients take antihypertensive drugs and 35 patients who have a stroke history from the registration book, which is arranged based on individual Medical Record Number (MRN) by taking the list of all adult patients in the medical register as a sampling frame. The interval (K) was calculated for cases and controls separately. Therefore, the interval (k) for cases was every 9, and (k) for controls was every ten (10) intervals. Lottery methods were used from one to nine for cases and from one to ten for controls to select the first patient Medical Record Number (MRN). (S1 Figure in S1 File)

## Study variables

*Dependent variable:* was stroke with their respective codes, Cases =1 and Controls =0
*Independent variable*

**Socio-demographic factors:** age, sex, marital status, occupation, residency and educational status, Family history of CVD.

**Lifestyle & Behavioral factors:** Physical exercise, Smoking, alcohol.

**Dietary related factors:** excessive salt in diet, fatty food use.

**Comorbidities:** Hypertension, Diabetes mellitus, Dyslipidemia, Obesity, and Cardiac disease.

## Operational definition

- **Physically active:** if patients make regular physical activities of 30 minutes for 5 days and above per week [1].

- **Physically inactive** if patients are not engaged in regular exercise [1].

- **Alcohol drinker:** a person who consumes 10.5 units or more of alcohol each week [1].

- **Physical measurements and clinical factors**: Raised FBG> = 126 mg/dl, normal FBG < 126 mg/dl [1].

- **Cholesterol level**: high if 200 or more, and normal if less than 200 [1].

- **Body Mass Index (BMI)**: underweight (weight under 18.5), normal (18.5–24.9), overweight (weight 25–29.9), and obese (30 and beyond) [6].

- **Systolic blood pressure**: both in control (<140) and uncontrol ( > = 140) [19].

- **Diastolic blood pressure**: both in control (<90) and uncontrol ( > = 90) [19].

- **Dietary history** will include the regularity of intake of food items such as meat, green leafy vegetables, sugar, and salt at the table. Regular intake was defined as intake daily, weekly, or at least once per month versus none in a month [1].

- **Smoking status is** defined as a current smoker (individuals who smoked any tobacco in the past 12 months), never-smoker, or former smoker [1].

- **Family history**: of cardiovascular risk or diseases is defined as the self-reported history of hypertension, diabetes, dyslipidemia, stroke, cardiac disease, or obesity in a participant's father, mother, sibling, or second-degree relative [1].

## Data collection technique

Data were collected through interview-administered questionnaire techniques. Information on basic sociodemographic data of respondents and parents, socioeconomic status, dietary factors, the patient's comorbid status, and lifestyle factors for stroke was obtained from the patient or a close relative (for the unconscious participant) through a face-to-face interview. Data were abstracted by seven trained final-year Health officer (HO) students blinded to the study hypothesis using a standardized, paper data abstraction instrument in English.

The following measurements of blood pressure, height, and weight were made by data collectors during data collection: Weight was recorded using a calibrated United Nations Children's Fund (UNICEF) digital weighing scale while wearing light clothing and without shoes. Height was measured using an audiometer in centimeters in an upright position with a precision of 0.1 cm without shoes. For each person whose blood pressure we collected throughout the data collection, the average of two readings obtained five minutes apart was calculated using a mercury sphygmomanometer.

## Data quality control

Data quality was checked during questionnaire designing, data collection, and data entry. A questionnaire was pre-tested among 5% of study subjects at Boru Meda Hospital. The data collectors and supervisors were trained at the zonal town (Dessie) for one day on the objectives of the study and data quality. The questionnaire was prepared in English first and then translated into Amharic and back to English by language experts to check for consistency. The supervisors were checking daily for any incompleteness of the questionnaire.

## Data processing and analysis

The data were coded and entered using Epi-data Version 4.4.2 software. Data were exported to SPSS version 26 for further statistical analyses. Descriptive and analytical analyses were also performed. For continuous data, descriptive features were expressed as mean (standard deviation), median (interquartile range), and frequency distribution for categorical data. Frequency tables, graphs, and cross-tabulations were used to present the findings of the study. Multicollinearity among independent variables was checked based on variance inflation factor (VIF) or tolerance. Both bivariable and Multivariable logistic regression models were used to identify determinants of stroke among hypertension patients. Model fitness was checked using Hosmer-Lemshow statistics. The first bivariable analysis was made for each independent variable to the outcome variable, and those variables resulting in a p-value less than 0.25 were entered into the multivariable binary logistic regression model. In the final model, those variables with a p-value less than 0.05 were considered as statistically significant, and it was presented by adjusted odds ratio (AOR), with a 95% confidence level (CI) to show the strength and direction of the association.

### Research ethics approval

Ethical clearance was obtained from the Tropical College of Medicine Department of Nursing Ethical Review Committee (ERC) with a reference number TCOM/Res/1716/2023. Official permission was obtained from DCSH and Kombolcha General Hospital's chief executive director and study participants were informed about the purpose of the study. The information was collected after obtaining written informed consent from the participants. The freedom of research participants to leave the study at any time was respected. Throughout the study, confidentiality was guaranteed and the information was recorded anonymously. All methods were performed in compliance with the Declaration of Helsinki.

## Results

A total of 105 cases and 420 control participants were involved in the study yielding a response rate of 100%.

### Socio-economic and demographic characteristics

62 (58.9%) and 280 (66.6%) were males for cases and controls respectively. The mean age of cases was 55.23 years (SD ± 10.588) and 52.45 years (SD ± 9.45) for controls. Among the total participants, 85 (80.9%) cases and 123 (29.4%) controls were urban residents. Of the total respondents, 49 (46.3%) among cases and 200 (47.7%) among controls were married respectively. (S2 Table in S1 File)

### Lifestyle and behavioral characteristics

Among the participants, the majority of the cases (61.1%) and (37.7%) of controls have a family history of hypertension. Most of the cases (51.4%) and controls (71.7%) were not physically active and did not take part in regular physical exercise. Of the total respondents, 19 (17.7%) among cases and 73 (17.4%) among controls were underweight respectively. Eighteen percent of the respondents had less than 1500 Birr monthly income in cases while controls had 18.3% less than 1500 Birr. (S3 Table in S1 File)

### Comorbid characteristics

Among the total participants, 57.1% of cases and 36.3% of controls were living with any type of chronic non-communicable disease (NCD). Most of the cases (51.4%) and controls (71.4%) have hypertension disease for more than five years duration. (S4 Table in S1 File)

### Dietary characteristics

Patients who had monthly consumption of fatty foods both in cases and controls were 50.3% and 48.3% respectively. Study participants from cases 46 (44%) and controls 164 (39.1%) experienced consumption of salty foods daily. Patients who had a consumption of vegetables daily both in cases and controls were 24% and 12.9% respectively. (S5 Table in S1 File)

### Determinants of stroke among hypertension patients

To ascertain the presence of an association between the dependent variable and the independent variables at (P 0.05) level of significance, bivariable and multivariable logistic regression models were fitted. In the bivariate analysis, variables that had a P-value of less than 0.25 with stroke among hypertension patients were used in the multivariable logistic regression analysis.

In the final multivariable logistic regression model, variables such as family history of hypertension (AOR: 2.8, 95% CI: 1.8-4.3), physically inactive (AOR: 2.13, 95% CI: 1.4-3.3), current smoking cigarettes (AOR: 3.2, 95% CI: 2.1-5.2), current alcohol drinking (AOR: 4.7, 95% CI: 2.87-7.9) and having heart disease (AOR: 2.04, 95% CI: 1.3-3.25). (S6 Table in S1 File)

## Discussion

The present study identifies certain determinant factors among adult patients with stroke compared to hypertension without stroke patients. This study found that stroke patients who have a family history of hypertension were 2.8 times more likely to develop stroke than their counterparts. This finding is consistent with studies conducted in southeast Ethiopia [8]. But inconsistent with studies done in Tanzania [22]. The possible reasons for the similarities might be due to blood relatives tend to have many of the same genes that can predispose a person to high blood pressure and, thereby stroke. Combining genetics with bad lifestyle habits like smoking and eating an unhealthy diet might make the risk of stroke even higher. The difference might be due to differences in socio-demographic status.

The findings of this study indicated that patients who smoke cigarettes currently were significantly associated with or 3.2 times more likely to develop stroke. This finding is consistent with studies done in Ghana [4], Nigeria [21], and Felegehiwot referral hospital [2]. The similarities might be due to smoking reducing the levels of good cholesterol (also called high-density lipoprotein, HDL) in the bloodstream and increasing levels of bad cholesterol (also called low-density lipoprotein, LDL). Thus, having low levels of good cholesterol in the body increases the risk of stroke.

This study indicated that patients with physical inactivity were 2.13 times more likely to develop strokes than their counterparts. This was inconsistent with studies done in Nigeria [21], and Uganda [23]. But consistent with studies done in Ghana [4]. The possible reasons for the differences might be lifestyle behavior, and the similarities might be that physical inactivity is a strong risk factor for stroke despite an increased burden of other cardiovascular disease risk factors. Being sedentary might cause fat to accumulate in the arteries. A heart attack may result from injury or clogging (burst) of the arteries supplying blood to the heart. Stroke may result if this occurs in the blood vessels that supply the brain.

This study found that previous and current consumption of alcohol was significantly associated with, or 4.7 times more likely to cause stroke. This study was supported by studies done in Ayider Comprehensive Specialized Hospital in Tigray [1], and Felegehiwot Referral Hospital in Bahir Dar [2]. The possible reasons for the similarities might be that there are several ways in which alcohol consumption may increase the risk of stroke by increasing patients' blood pressure and weight (overweight and obesity). Beyond this, it may increase the risk of diabetes, liver damage, and atrial fibrillation.

The findings of this study indicated that patients with heart disease were 2.04 times more likely to develop stroke than their counterparts. This was inconsistent with studies done in Nigeria [21] and Bahir-dar [2]. The possible reasons for the differences might be due to differences in socioeconomic status and lifestyle changes. Additionally, common heart conditions can make patients more susceptible to strokes. For instance, coronary artery disease raises your risk for stroke because the arteries become clogged with plaque, which prevents oxygen-rich blood from reaching the brain. Blood clots may develop as a result of various heart conditions, including heart valve problems, irregular heartbeats (including atrial fibrillation), and enlarged heart chambers.

### Strength and limitation of the study

- Being a hospital-based study and as such the finding cannot represent the general population was the limitation of this study.

- However, one of its key strengths is that it makes a significant contribution to the body of knowledge regarding the risk factors for stroke among hypertensive patients.

### Conclusion

Among hypertensive patients, the modifiable factors associated with stroke occurrence included having a family history of hypertension, cigarette smoking, physical inactivity, alcohol consumption, and heart disease was positively associated with the occurrence of stroke in Dessie Comprehensive Specialized Hospital and Kombolcha Primary Hospital. It's recommended that health institutions should put strategies for screening and management of early hypertension and other cardiovascular diseases for the reduction of stroke prevalence and incidence. Nurses and health extension professionals should give community IEC activities to create awareness about the importance of improving lifestyle and personal behavior (like avoiding alcohol intake and smoking) to prevent the incidence of stroke.

### Supporting Information

**S1 File. "Combined figures and table showing schematic presentation of sampling technique, sample size calculation, socio-demographic characteristics, lifestyle and behavioral factors, comorbid illnesses of adult stroke patients, dietary factors of adult stroke patients and bivariate and multivariable logistic regression model.**
(DOCX)

### Acknowledgment

We would like to first and foremost express our sincere gratitude to Tropical College of Medicine for providing this opportunity. We would also like to express our gratitude to the Dessie Comprehensive Specialized Hospital and Kombolcha General Hospital for helping us gather data for this thesis.

### Author contributions

**Conceptualization:** Muluken Demile, Molla Kassie Yenesew.

**Data curation:** Muluken Demile, Molla Kassie Yenesew.

**Formal analysis:** Muluken Demile, Molla Kassie Yenesew.

**Funding acquisition:** Muluken Demile.

**Investigation:** Muluken Demile, Molla Kassie Yenesew.

**Methodology:** Yonas Fissha Adem, Muluken Demile, Molla Kassie Yenesew, Abebe Bekele Jibat.

**Project administration:** Muluken Demile, Molla Kassie Yenesew, Abebe Bekele Jibat.

**Resources:** Muluken Demile, Abebe Bekele Jibat.

**Software:** Abebe Bekele Jibat.

**Supervision:** Muluken Demile, Abebe Bekele Jibat.

**Validation:** Muluken Demile, Endalew Minwuye Andargie.

**Visualization:** Muluken Demile, Dagnachew Melak, Endalew Minwuye Andargie.

**Writing – original draft:** Yonas Fissha Adem, Yohannes Kibret, Dagnachew Melak, Endalew Minwuye Andargie.

**Writing – review & editing:** Yonas Fissha Adem, Yohannes Kibret, Dagnachew Melak, Endalew Minwuye Andargie.

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
