## [Decision Letter · Decision Letter 0]

15 Sep 2024

PONE-D-24-27441Determinants of Stroke among Adult Hypertension Patients in Dessie Comprehensive Specialized Hospital and Kombolcha General Hospital, Amhara, Ethiopia: A Case-Control StudyPLOS ONE

Dear Dr. Adem,

Thank you for submitting your manuscript to PLOS ONE. After careful consideration, we feel that it has merit but does not fully meet PLOS ONE’s publication criteria as it currently stands. Therefore, we invite you to submit a revised version of the manuscript that addresses the points raised during the review process.

**ACADEMIC EDITOR (comments)**

This manuscript requires substantial improvement in clarity, coherence, and conciseness. The methods section needs thorough revision. please see the below details of comments for each section in the study:

Introduction

The introduction needs to be revised. The rationale for the study is not well articulated

Some phrases and information are not clearly stated, some are repeated or redundant.

Authors need to ensure that each section flows logically into the next for better readability.

Authors need to provide more actionable recommendations based on their findings.

The entire manuscript writings needs improvement for ease of understanding and authors may need to seek the services of professional English language editors.

Abbreviation is not appropriate used

Some parts contain redundancy or could be better structured to improve readability.

Authors need to ensure each paragraph transitions smoothly to the next.

It would be more effective if the introduction flowed from the background to the specific problem being addressed.

Methods

The statistical methods should be rewritten

Result section

This is not well articulated and needs improvement which needs to be interpreted based on the findings.

Conclusion

The conclusion lacks clarity and need to be based on the study. Please rewrite this section.

We look forward to receiving your revised manuscript.

Kind regards,

Denekew Bitew Belay, Ph.D

Academic Editor

PLOS ONE

3. In this instance it seems there may be acceptable restrictions in place that prevent the public sharing of your minimal data. However, in line with our goal of ensuring long-term data availability to all interested researchers, PLOS’ Data Policy states that authors cannot be the sole named individuals responsible for ensuring data access (http://journals.plos.org/plosone/s/data-availability#loc-acceptable-data-sharing-methods).

4. We notice that your supplementary figures are uploaded with the file type 'Figure'. Please amend the file type to 'Supporting Information'. Please ensure that each Supporting Information file has a legend listed in the manuscript after the references list.

5. We notice that your supplementary tables are included in the manuscript file. Please remove them and upload them with the file type 'Supporting Information'. Please ensure that each Supporting Information file has a legend listed in the manuscript after the references list.

Reviewers' comments:

Reviewer's Responses to Questions

**Comments to the Author**

1. Is the manuscript technically sound, and do the data support the conclusions?

Reviewer #1: Partly

Reviewer #2: Partly

Reviewer #3: Yes

2. Has the statistical analysis been performed appropriately and rigorously? 

Reviewer #1: I Don't Know

Reviewer #2: Yes

Reviewer #3: Yes

3. Have the authors made all data underlying the findings in their manuscript fully available?

Reviewer #1: No

Reviewer #2: Yes

Reviewer #3: Yes

4. Is the manuscript presented in an intelligible fashion and written in standard English?

Reviewer #1: No

Reviewer #2: No

Reviewer #3: Yes

5. Review Comments to the Author

Reviewer #1: Dear Authors,

Thank you for your dedication to researching the important topic of "Determinants of Stroke among Adult Hypertension Patients in Dessie Comprehensive Specialized Hospital and Kombolcha General Hospital, Amhara, Ethiopia: A Case-Control Study."

While I acknowledge the effort you have put into this study, it is essential to address certain areas that require further attention to enhance the objectivity and precision of the findings. Specifically, I recommend additional quantification of the identified significant factors to reduce subjectivity and enhance the robustness and reliability of the research outcomes. Additionally, I suggest that the conclusions drawn should be firmly rooted in the empirical evidence presented in the study results to uphold the validity and reliability of the research findings.

Reviewer #2: 1. Several study is conducted in Ethiopia For example: https://biomedres.us/pdfs/BJSTR.MS.ID.008576.pdf, https://doi.org/10.1186/s40001-023-01192-6, https://doi.org/10.1371/journal.pone.0228650, https://doi.org/10.2147/DNND.S435806. What's new in your research?

2. I would suggest the authors to find a native English speaker for English editing.

3. Please describe why the independent variables you chose are relevant and how they were defined.

4. In the introduction, the authors have pointed out the deficiencies of the previous studies. The authors are expected to explain in what way the findings of this study were different from the previous studies.

5. Which populations are your sources and studies? Are those patients with hypertension and stroke patients, or just those with hypertension?

6. Delete the passage from lines 259–262 on page 22.

7. How can multicollnarity be tested?

8. The outcome variable needs to be well specified.

9. Did you prepare the table using standard form?

Reviewer #3: Thank you for giving me the opportunity to review the manuscript titled “Determinants of Stroke among Adult Hypertension Patients in Dessie Comprehensive Specialized Hospital and Kombolcha General Hospital, Amhara Ethiopia: A Case-Control Study’ I apologize for any inconvenience caused by the delay in my review.

The problem raised by the researchers and the hospital data used are impressive and make the work suitable for publication in PLoS ONE. However, the manuscript has several weaknesses that need addressing: Addressing the following issues will enhance the clarity and professionalism of the manuscript, making it more suitable for publication.

1. Language and Editing: The manuscript requires significant improvement in language editing. This includes correcting spelling and typographical errors, as well as grammatical mistakes.

2. Abbreviations: Ensure all abbreviations are properly defined when first introduced. For instance:

MRN (line 153), BMI (line 174), HO (line 191), UNICEF (line 194), "Patent" should be corrected to "patient" (line 258)

Generally, please make a thorough review of the document to correct these errors and ensure consistency throughout.

Introduction

The introduction is relatively well written

Methods and materials

1. "Study Design, Area, and Period": The subtitle "Study Design, Area, and Period" may not clearly convey the intended content. Consider rephrasing it to better reflect the specifics of your study, such as "Study Design, Geographic Scope, and Time Frame," to make it more descriptive and accurate.

2. Source of Information: Clarify the source of your information, especially regarding the figure of 10 million in the DCSH catchment areas. Verify this number and cite the source accurately. If you cannot provide a source, consider removing this figure from the document to maintain credibility.

3. Source Population vs. Study Population: Define the difference between the source population and the study population. If your intent is to describe a specific group from which the sample is drawn, use "study population" consistently. Avoid confusion by choosing one term and using it throughout the document.

4. Sample Size Determination: The explanation of your sample size determination is unclear. Provide a detailed description and support it with appropriate citations or sources. This will help validate your methodology and ensure that the sample size is justified based on relevant statistical or research standards.

5. Operational Definitions: The operational definitions are well-written and supported with evidence. Ensure that each definition is clearly defined and referenced with appropriate sources to enhance the reliability and validity of your study.

These adjustments should help clarify and strengthen the presentation of your study's methodology and findings.

Data quality control

Is there a specific reason why the pre-test is being conducted at Boru Meda Hospital, which is another hospital included in your study?

Data processing and analysis: What exactly does this entail? Are you referring to statistical methods, or something else? Please revise the methodology section, briefly and in detail, to clarify how you could apply bivariable and multivariable logistic regression. Since these methods are used to compare models for calculating your Crude Odds Ratio (COR) and Adjusted Odds Ratio (AOR), please also include the relevant statistical model equations in your explanation.

Results:

In Table 1, for the "Current Occupation" category, you classify occupations as private, governmental, and NGOs. However, it's unclear where groups like unemployed individuals, farmers, or merchants fall within these classifications. Could you clarify which category they belong to or consider adding additional categories for these occupations?

Additionally, please spell out "Non-Governmental Organizations (NGOs)" when first mentioned, and avoid using "gov’tal" as it is not a proper abbreviation. Instead, use "governmental" for clarity.

The descriptive statistics provided from Table 1 to Table 4 are insightful, but they require more in-depth explanations. Ensure that each table's findings are clearly explained to highlight the important trends or patterns observed in the data.

As for Table 5, which presents the analysis of your logistic regression models, while the presentation is good, it would benefit from more detailed interpretations. Specifically, for the significant variables, please interpret the odds ratios (OR) of each category in relation to their reference groups. For example, explain how an odds ratio above or below 1 indicates increased or decreased likelihood of the outcome compared to the reference category. This will help clarify the real-world implications of the results.

Discussion and conclusion

The discussion section of the manuscript is relatively well written, providing a good analysis of the findings. However, the conclusion requires improvement to more effectively summarize the key results and their implications.

Recommendations: The recommendations should be more specific. Clearly identify the target audience for these recommendations, whether they are intended for national, regional, zonal, or woreda levels, or any other specific bodies. This specificity will help ensure that the recommendations are actionable and relevant to the appropriate stakeholders.

6. PLOS authors have the option to publish the peer review history of their article (what does this mean? ). If published, this will include your full peer review and any attached files.

**Do you want your identity to be public for this peer review?** For information about this choice, including consent withdrawal, please see our Privacy Policy .

Reviewer #1: No

Reviewer #2: **Yes: ** Setegn Muche Fenta

Reviewer #3: No

---

## [Author Response · Author response to Decision Letter 1]

11 Dec 2024

Date: 10 /10/2024

Denekew Bitew Belay

PLOS ONE

Dear… Denekew Bitew Belay

Thank you for giving us the opportunity to submit a revised draft of our manuscript titled

“Determinants of Stroke among Adult Hypertension Patients in Dessie Comprehensive Specialized Hospital and Kombolcha General Hospital, Amhara, Ethiopia: A Case-Control Study” to the PLOS ONE. We appreciate the time and effort that you and the reviewers have dedicated to providing your valuable feedback on our manuscript. We are grateful to the reviewers for their insightful comments on our paper. We have been able to incorporate changes to reflect most of the suggestions provided by the reviewers. We have highlighted the changes within the manuscript.

Please let us know if you still have any questions or concerns about the manuscript. We will be happy to address them, now in a timely manner.

Sincerely,

Yonas Fissha Adem

Point by Point Response to - Editor

Point by Point Response to – Reviewers

Academic Editor

This manuscript requires substantial improvement in clarity, coherence, and conciseness. The methods section needs thorough revision. please see the below details of comments for each section in the study:

Introduction

The introduction needs to be revised. The rationale for the study is not well articulated

Some phrases and information are not clearly stated, some are repeated or redundant.

Authors need to ensure that each section flows logically into the next for better readability.

Authors need to provide more actionable recommendations based on their findings.

The entire manuscript writings needs improvement for ease of understanding and authors may need to seek the services of professional English language editors.

Abbreviation is not appropriate used

Some parts contain redundancy or could be better structured to improve readability.

Authors need to ensure each paragraph transitions smoothly to the next.

It would be more effective if the introduction flowed from the background to the specific problem being addressed.

Response: We would like to thank the dear editorial for your frank comment. We reviewed and made extensive adjustments in the revised manuscript.

Methods

The statistical methods should be rewritten

Response: We appreciate the suggestion. Changes were made to the method part of revised manuscript.

Result section

This is not well articulated and needs improvement which needs to be interpreted based on the findings.

Response: We appreciate your valuable suggestion. We have made improvement in the revised manuscript as per your expectations.

Conclusion

The conclusion lacks clarity and need to be based on the study. Please rewrite this section.

Response: We appreciate your thoughtful observation regarding this issue. We've made changes to this section based on your comments.

Reviewers' comments:

Reviewer's Responses to Questions

Comments to the Author

Reviewer #1: Dear Authors,

Thank you for your dedication to researching the important topic of "Determinants of Stroke among Adult Hypertension Patients in Dessie Comprehensive Specialized Hospital and Kombolcha General Hospital, Amhara, Ethiopia: A Case-Control Study."

While I acknowledge the effort you have put into this study, it is essential to address certain areas that require further attention to enhance the objectivity and precision of the findings. Specifically, I recommend additional quantification of the identified significant factors to reduce subjectivity and enhance the robustness and reliability of the research outcomes. Additionally, I suggest that the conclusions drawn should be firmly rooted in the empirical evidence presented in the study results to uphold the validity and reliability of the research findings.

Response: Thank you for this important observation. As much as possible we have tried to improve and update the content of the manuscript as per your recommendation.

Reviewer #2: 1. Several study is conducted in Ethiopia For example: https://biomedres.us/pdfs/BJSTR.MS.ID.008576.pdf, https://doi.org/10.1186/s40001-023-01192-6, https://doi.org/10.1371/journal.pone.0228650, https://doi.org/10.2147/DNND.S435806. What's new in your research?

Response: dear reviewer, thanks for your valuable comment. while the number of stroke patients being admitted to hospitals as a result of hypertension is rising, few results attempt to investigate those factors through case-control studies since these few studies concentrate more on the magnitude not for the factors. Furthermore, there is no similar study conducted in the study area, that accounts for differences in socioeconomic status, and access to treatment and it is useful for designing contextual interventions.

2. I would suggest the authors to find a native English speaker for English editing.

Response: Thank you for your careful inspection. Done as requested.

3. Please describe why the independent variables you chose are relevant and how they were defined.

Response: dear reviewer, thank you for pointing this out. The independent variables were taken from different literature which might be a factor for outcome variable. Some independent variables were defined in the operational definition of this study.

4. In the introduction, the authors have pointed out the deficiencies of the previous studies. The authors are expected to explain in what way the findings of this study were different from the previous studies.

Response: Thank you for your thoughtful comment. We have already explained it above.

5. Which populations are your sources and studies? Are those patients with hypertension and stroke patients, or just those with hypertension?

Response: We appreciate the reviewer’s important opinion. All adult (>18 years of age) hypertensive patients who were on follow-up in DCSH and Kombolcha General Hospital visit during the data collection period were the study population which were the actual sample size was drawn, this population is called sample population. But All adult hypertensive patients and stroke patients from hypertension backgrounds in DCSH and Kombolcha General Hospital visit the facilities were the source population which finally this study result was generalized. We have made it short and precise in the revised manuscript.

6. Delete the passage from lines 259–262 on page 22.

Response: Thank you very much for your comments. But this passage is important because this passage talk about how bivariable logistic analysis was done then the candidate is chosen for final multivariable logistic analysis. So, we considered this passage is essential but if it is mandatory, it’s easy to delete it.

7. How can multicollinearity be tested?

Response: Thank you for your insightful comment. We have incorporated your idea in the data analysis part of a revised manuscript.

8. The outcome variable needs to be well specified.

Response: We thank you for your valuable opinion. Our outcome variable is all sampled adult stroke patients older than eighteen years were from a hypertension background diagnosed by the neurologist (consultant internist) as cases and all adult hypertension patients older than eighteen years who did not exhibit clinical signs of stroke or a history of stroke as controls. It’s placed in short in the revised manuscript.

9. Did you prepare the table using standard form?

Response: Thank you for the comments. We have made adjustments accordingly.

Reviewer #3: Thank you for giving me the opportunity to review the manuscript titled “Determinants of Stroke among Adult Hypertension Patients in Dessie Comprehensive Specialized Hospital and Kombolcha General Hospital, Amhara Ethiopia: A Case-Control Study’ I apologize for any inconvenience caused by the delay in my review.

The problem raised by the researchers and the hospital data used are impressive and make the work suitable for publication in PLoS ONE. However, the manuscript has several weaknesses that need addressing: Addressing the following issues will enhance the clarity and professionalism of the manuscript, making it more suitable for publication.

1. Language and Editing: The manuscript requires significant improvement in language editing. This includes correcting spelling and typographical errors, as well as grammatical mistakes.

Response: dear reviewer, we agreed with the reviewer's comment. Done as requested.

2. Abbreviations: Ensure all abbreviations are properly defined when first introduced. For instance:

MRN (line 153), BMI (line 174), HO (line 191), UNICEF (line 194), "Patent" should be corrected to "patient" (line 258)

Generally, please make a thorough review of the document to correct these errors and ensure consistency throughout.

Response: dear reviewer, thank you for your valuable feedback. We have corrected it accordingly in the revised manuscript.

Introduction

The introduction is relatively well written

Response: dear reviewer, thank you very much for your motivational comment.

Methods and materials

1. "Study Design, Area, and Period": The subtitle "Study Design, Area, and Period" may not clearly convey the intended content. Consider rephrasing it to better reflect the specifics of your study, such as "Study Design, Geographic Scope, and Time Frame," to make it more descriptive and accurate.

Response: dear reviewer, thank you for spotting this. It has been rewritten to give clarity in the revised manuscript.

2. Source of Information: Clarify the source of your information, especially regarding the figure of 10 million in the DCSH catchment areas. Verify this number and cite the source accurately. If you cannot provide a source, consider removing this figure from the document to maintain credibility.

Response: Thank you for your thoughtful concern of our study. We had removed this figure from the revised manuscript.

3. Source Population vs. Study Population: Define the difference between the source population and the study population. If your intent is to describe a specific group from which the sample is drawn, use "study population" consistently. Avoid confusion by choosing one term and using it throughout the document.

Response: Thank you for your detailed comments. We have revised the manuscript according to your comments.

4. Sample Size Determination: The explanation of your sample size determination is unclear. Provide a detailed description and support it with appropriate citations or sources. This will help validate your methodology and ensure that the sample size is justified based on relevant statistical or research standards.

Response: Thank you so much for your careful check. But we have already explained it as much as possible even by putting a reference from previous study. But if you have a specific comments, we are happy to modify it.

5. Operational Definitions: The operational definitions are well-written and supported with evidence. Ensure that each definition is clearly defined and referenced with appropriate sources to enhance the reliability and validity of your study. These adjustments should help clarify and strengthen the presentation of your study's methodology and findings.

Response: Thank you very much for your good comment.

Data quality control

Is there a specific reason why the pre-test is being conducted at Boru Meda Hospital, which is another hospital included in your study?

Response: Thank you for your rigorous comment. pretesting is designed to make sure that people understand the questions, and that there isn’t anything in the data that indicates that the information is inaccurate. Pretest is also important for budget allocation. But pretest is done for primary data in different facility other than the study area. Because if pretest is done on same area for primary data the information linkage is high so finally we have got biased information from respondents. But for secondary data we can done pretest on the same area.

Data processing and analysis: What exactly does this entail? Are you referring to statistical methods, or something else? Please revise the methodology section, briefly and in detail, to clarify how you could apply bivariable and multivariable logistic regression. Since these methods are used to compare models for calculating your Crude Odds Ratio (COR) and Adjusted Odds Ratio (AOR), please also include the relevant statistical model equations in your explanation.

Response: Thank you for your valuable input. We modified it accordingly in the revised manuscript.

Results:

In Table 1, for the "Current Occupation" category, you classify occupations as private, governmental, and NGOs. However, it's unclear where groups like unemployed individuals, farmers, or merchants fall within these classifications. Could you clarify which category they belong to or consider adding additional categories for these occupations?

Response: Thank you for your reminder. Sorry for making us a mistake here and we have improved the contents in the revised manuscript. In this study we were consider private (self-employed) consists merchant and farmers. We have also forgotten “Other” options. We have made correct all this issue. Also we have done Bivariable analysis again and p-value became above 0.25 so it’s not a candidate for multivariable analysis as shown below.

Variables in the Equation

B S.E. Wald df Sig. Exp(B) 95% C.I.for EXP(B)

Lower Upper

Step 1a Occupation? .346 4 .987

Occupation?(1) .022 .415 .003 1 .958 1.022 .453 2.305

Occupation?(2) .313 .600 .272 1 .602 1.367 .422 4.430

Occupation?(3) .138 .441 .097 1 .755 1.148 .483 2.726

Occupation?(4) .130 .609 .046 1 .831 1.139 .345 3.759

Constant 1.191 .233 26.128 1 .000 3.292

a. Variable(s) entered on step 1: Occupation?.

Additionally, please spell out "Non-Governmental Organizations (NGOs)" when first mentioned, and avoid using "gov’tal" as it is not a proper abbreviation. Instead, use "governmental" for clarity.

Response: Thank you for your reminder. We have implemented the necessary modifications in the revised manuscript.

The descriptive statistics provided from Table 1 to Table 4 are insightful, but they require more in-depth explanations. Ensure that each table's findings are clearly explained to highlight the important trends or patterns observed in the data.

Response: Thank you for raising this important point. We have added the in-depth explanation based on your comment in the revised manuscript.

As for Table 5, which presents the analysis of your logistic regression models, while the presentation is good, it would benefit from more detailed interpretations. Specifically, for the significant variables, please interpret the odds ratios (OR) of each category in relation to their reference groups. For example, explain how an odds ratio above or below 1 indicates increased or decreased likelihood of the outcome compared to the reference category. This will help clarify the real-world implications of the results.

Response: Thank you for your suggestions. We have included the exact number of odds ratios in the revised manuscript.

Discussion and conclusion

The discussion section of the manuscript is relatively well written, providing a good analysis of the findings. However, the conclusion requires improvement to more effectively summarize the key results and their implications.

Response: We would like to thank the reviewer for her/his kind words regarding our paper. We have revised in the manuscript based on your comment,

Recommendations: The recommendations should be more specific. Clearly identify the target audience for these recommendations, whether they are intended for national, regional, zonal, or woreda levels, or any other specific bodies. This specificity will help ensure that the recommendations are actionable and relevant to the appropriate stakeholders.

Response: Thank you. We reviewed the recommendation and made it rewrite.

---

## [Decision Letter · Decision Letter 1]

2 Jan 2025

PONE-D-24-27441R1Determinants of Stroke among Adult Hypertension Patients in Dessie Comprehensive Specialized Hospital and Kombolcha General Hospital, Amhara, Ethiopia: A Case-Control StudyPLOS ONE

Dear Dr. Adem,

Thank you for submitting your manuscript to PLOS ONE. After careful consideration, we feel that it has merit but does not fully meet PLOS ONE’s publication criteria as it currently stands. Therefore, we invite you to submit a revised version of the manuscript that addresses the points raised during the review process.

We look forward to receiving your revised manuscript.

Kind regards,

Denekew Bitew Belay, Ph.D

Academic Editor

PLOS ONE

Additional Editor Comments:

The manuscript still needs substantial improvements. Please pay attention and make the required revisions. The Sample size determination still needs improvements.

Reviewers' comments:

Reviewer's Responses to Questions

**Comments to the Author**

1. If the authors have adequately addressed your comments raised in a previous round of review and you feel that this manuscript is now acceptable for publication, you may indicate that here to bypass the “Comments to the Author” section, enter your conflict of interest statement in the “Confidential to Editor” section, and submit your "Accept" recommendation.

Reviewer #2: All comments have been addressed

Reviewer #3: (No Response)

2. Is the manuscript technically sound, and do the data support the conclusions?

Reviewer #2: Yes

Reviewer #3: Partly

3. Has the statistical analysis been performed appropriately and rigorously? 

Reviewer #2: Yes

Reviewer #3: No

4. Have the authors made all data underlying the findings in their manuscript fully available?

Reviewer #2: Yes

Reviewer #3: Yes

5. Is the manuscript presented in an intelligible fashion and written in standard English?

Reviewer #2: Yes

Reviewer #3: Yes

6. Review Comments to the Author

Reviewer #2: I appreciate the editor giving me the opportunity to assess this work, and the authors answered all of my questions.

Reviewer #3: Dear Author/s

Thank you for addressing many of the comments and making improvements in the manuscript. However, there are still some concerns that were not adequately addressed in this round of revisions.

1. The abbreviations are not correctly abbreviated eg. Medical Record Number (MRN) mentioned two times on page 8 and 9.

2. Some of my questions are not replied, for example

a. my suggestions on sample size determination “Sample Size Determination: The explanation of your sample size determination is unclear. Provide a detailed description and support it with appropriate citations or sources. This will help validate your methodology and ensure that the sample size is justified based on relevant statistical or research standards. “

The Response is “Thank you so much for your careful check. But we have already explained it as much as possible even by putting a reference from previous study. But if you have a specific comment, we are happy to modify it.”

Now I can ask more questions as What is sample size determinations and what is it is purpose? Sample size determination is the process of calculating the number of participants or observations needed in a study to achieve reliable and valid results. Therefore, where is your sample size calculations? how can you determine the sample size? Your readers need clarity about your sample size determination? If your sample size determination is wrong then what will happen about work?

b. My question “Data processing and analysis: What exactly does this entail? Are you referring to statistical methods, or something else? Please revise the methodology section, briefly and in detail, to clarify how you could apply bivariable and multivariable logistic regression. Since these methods are used to compare models for calculating your Crude Odds Ratio (COR) and Adjusted Odds Ratio (AOR), please also include the relevant statistical model equations in your explanation.”

The response is: “Thank you for your valuable input. We modified it accordingly in the revised manuscript.” But I could not find the modified methods in the revised manuscript.

3. Under section: Study Design, Geographic Scope, and Time Frame

There is no clear evidence for your information, please put the references where did you get this information

4. In general, I could not identify where the responses to the comments are located, as there are no page or line numbers provided. Many answers are vague, using phrases like “we have done based on the comments” or “modified as per the comments,” without offering concrete evidence or detailed justifications. This lack of clarity makes it difficult to understand the rebuttals or acceptances in the manuscript.

Therefore, I strongly recommend providing clearer and more detailed explanations for the questions, including the specific page and line numbers where your responses can be found.

7. PLOS authors have the option to publish the peer review history of their article (what does this mean? ). If published, this will include your full peer review and any attached files.

**Do you want your identity to be public for this peer review?** For information about this choice, including consent withdrawal, please see our Privacy Policy .

Reviewer #2: No

Reviewer #3: No

---

## [Author Response · Author response to Decision Letter 2]

18 Jan 2025

Date: 03 /01/2025

Denekew Bitew Belay

PLOS ONE

Dear… Denekew Bitew Belay

Thank you for giving us the opportunity to submit a revised draft of our manuscript titled

“Determinants of Stroke among Adult Hypertension Patients in Dessie Comprehensive Specialized Hospital and Kombolcha General Hospital, Amhara, Ethiopia: A Case-Control Study” to the PLOS ONE. We appreciate the time and effort that you and the reviewers have dedicated to providing your valuable feedback on our manuscript. We are grateful to the reviewers for their insightful comments on our paper. We have been able to incorporate changes to reflect most of the suggestions provided by the reviewers. We have highlighted the changes within the manuscript.

Please let us know if you still have any questions or concerns about the manuscript. We will be happy to address them, now in a timely manner.

Sincerely,

Yonas Fissha Adem

Point by Point Response to - Editor

Point by Point Response to – Reviewers

Additional Editor Comments:

The manuscript still needs substantial improvements. Please pay attention and make the required revisions. The Sample size determination still needs improvements.

Response: Thank you for your suggestions. We will improve it as per your recommendations.

Reviewer #3: Dear Author/s

Thank you for addressing many of the comments and making improvements in the manuscript. However, there are still some concerns that were not adequately addressed in this round of revisions.

1. The abbreviations are not correctly abbreviated eg. Medical Record Number (MRN) mentioned two times on page 8 and 9.

Response: Thank you for your suggestions. But in our health facility setting MRN stands for medical record number. If you have other idea for MRN abbreviation. Give us your idea and we can modify according to your comments.

2. some of my questions are not replied, for example

A. my suggestions on sample size determination “Sample Size Determination: The explanation of your sample size determination is unclear. Provide a detailed description and support it with appropriate citations or sources. This will help validate your methodology and ensure that the sample size is justified based on relevant statistical or research standards. “

The Response is “Thank you so much for your careful check. But we have already explained it as much as possible even by putting a reference from previous study. But if you have a specific comment, we are happy to modify it.”

Now I can ask more questions as What is sample size determinations and what is it is purpose? Sample size determination is the process of calculating the number of participants or observations needed in a study to achieve reliable and valid results. Therefore, where is your sample size calculations? How can you determine the sample size? Your readers need clarity about your sample size determination? If your sample size determination is wrong then what will happen about work?

Response: Thank you very much dear reviewer, we have calculated the required sample size by using double population proportion formula using Epi Info 7 Software version 2.1.1. We didn’t use single population formula to get the largest sample size because this study isn’t cross-sectional study rather it’s a case control study so we calculate the sample size using factors associated with stroke from previous studies. We will show these factors using a table in the revised manuscript. You can check these using Epi info 7 software versions. Page 8 and line number 138-149

B. My question “Data processing and analysis: What exactly does this entail? Are you referring to statistical methods, or something else? Please revise the methodology section, briefly and in detail, to clarify how you could apply bivariable and multivariable logistic regression. Since these methods are used to compare models for calculating your Crude Odds Ratio (COR) and Adjusted Odds Ratio (AOR), please also include the relevant statistical model equations in your explanation.”

The response is: “Thank you for your valuable input. We modified it accordingly in the revised manuscript.” But I could not find the modified methods in the revised manuscript.

Response: Thank you for your suggestions. Dear, l seems that l have changed very well the “Data processing and analysis part” in previous revised manuscript as per your recommendation. But you are not satisfied. I will revise again as much as I can but l am stressed how I satisfied you. Because l considered that l was write all things about logistic regression. Page 11 and line number 218-223.

3. Under section: Study Design, Geographic Scope, and Time Frame

There is no clear evidence for your information, please put the references where did you get this information

Response: Thank you for your suggestions. We incorporated the reference for the information. page 6, line number 118

4. In general, I could not identify where the responses to the comments are located, as there are no page or line numbers provided. Many answers are vague, using phrases like “we have done based on the comments” or “modified as per the comments,” without offering concrete evidence or detailed justifications. This lack of clarity makes it difficult to understand the rebuttals or acceptances in the manuscript.

Therefore, I strongly recommend providing clearer and more detailed explanations for the questions, including the specific page and line numbers where your responses can be found.

Response: Thank you for your suggestions. We incorporated the page number and line numbers for each comments.

---

## [Editor Report · Decision Letter 2]

27 Jan 2025

Determinants of Stroke among Adult Hypertension Patients in Dessie Comprehensive Specialized Hospital and Kombolcha General Hospital, Amhara, Ethiopia: A Case-Control Study

PONE-D-24-27441R2

Dear Dr. Adem,

We’re pleased to inform you that your manuscript has been judged scientifically suitable for publication and will be formally accepted for publication once it meets all outstanding technical requirements.

Kind regards,

Denekew Bitew Belay, Ph.D

Academic Editor

PLOS ONE
---

## [Editor Report · Acceptance letter]

PONE-D-24-27441R2

PLOS ONE

Dear Dr. Adem,

I'm pleased to inform you that your manuscript has been deemed suitable for publication in PLOS ONE. Congratulations! Your manuscript is now being handed over to our production team.

Kind regards,

on behalf of

Dr. Denekew Bitew Belay

Academic Editor

PLOS ONE